# Antibiotic Stewardship—Twenty Years in the Making

**DOI:** 10.3390/antibiotics8010007

**Published:** 2019-01-24

**Authors:** Esmita Charani, Alison Holmes

**Affiliations:** National Institute for Health Research Health Protection Research Unit in Healthcare Associated Infections and Antimicrobial Resistance, 8th Floor Commonwealth Building, Imperial College London, London W12 0HS, UK; alison.holmes@imperial.ac.uk

**Keywords:** antibiotic stewardship, culture, social science, antimicrobial resistance

## Abstract

In the last 20 years, efforts were made to optimize antibiotic use in hospitals across the world as a means of addressing the increasing threat of antibiotic resistance. Despite robust evidence supporting optimal practice, antibiotic decision-making remains sub-optimal in many settings, including in hospitals. Globally, resources remain a limiting factor in the implementation of antibiotic stewardship programs. In addition, antibiotic decision-making is a social process dependent on cultural and contextual factors. Cultural boundaries in healthcare and across specialties still limit the involvement of allied healthcare professionals in stewardship interventions. There is variation in the social norms and antibiotic-prescribing behaviors between specialties in hospitals. The cultural differences between specialties and healthcare professionals (1) shape the shared knowledge within and across specialties in the patient pathway, and (2) result in variation in care, thus impacting patient outcomes. Bespoke stewardship interventions that account for contextual variation in practice are necessary.

## 1. Introduction

The first time the term “stewardship” was used in relation to antibiotic use was in an article in New Horizons in 1996 [1]. The authors called for an urgent need to address the then reported growing problem of antimicrobial resistance (AMR) in hospitals and concluded that more evidence in the shape of large-scale trials were required to establish how best to control the problem and “optimize antimicrobial use ‘stewardship’” [1]. The following year, the Society for Healthcare Epidemiology of America and Infectious Diseases Society of America (IDSA) published guidelines for the prevention of AMR in hospitals [2]. These guidelines set out for the first time the criteria for applied infection control programs in hospitals. The recommended criteria included (1) a system for monitoring bacterial resistance and antibiotic usage; (2) developing practice guidelines for the control and use of antibiotics; (3) adopting the Center for Disease Control and Prevention (CDC) Guidelines for Isolation Precautions in Hospitals; (4) utilizing hospital committees to develop local policies; (5) making hospital administration accountable for the implementation and enforcement of policies adopted by the hospital committees; and (6) measuring outcomes to evaluate the effectiveness of policies put in place [2]. The authors define appropriate antibiotic stewardship programs (ASPs) to be synonymous with “antibiotic control” and go on to say that appropriate antibiotic stewardship should include “not only the limitation of use of inappropriate agents, but also the appropriate selection, dosing, and duration of antibiotic therapy to achieve optimal efficacy in managing infections. The authors also highlighted the relevance of a multidisciplinary team approach and defined the contribution of pharmacists and administrators in the development of an optimal “antibiotic control program”. The premise of these seminal stewardship guidelines was based on a two-pronged approach for preventing transmission of infection through infection prevention and control (IP&C) and the optimization of antibiotic use. From then on, antibiotic stewardship was increasingly used to define the host of interventions from education and training, to decision support systems, restriction, and audit and feedback, which can be incorporated into the strategic agenda of organizations seeking to optimize antibiotic use. The effectiveness of these interventions was assessed through peer-reviewed publications and several systematic reviews [3,4,5]. At the same time, antimicrobial resistance (AMR) continued to relentlessly spread and become increasingly more difficult to treat and contain [6,7]. Interestingly, the two-pronged approach of IP&C and antibiotic use optimization diverged into two separate movements in the fight against AMR, because of ASP. Twenty years passed since antibiotic stewardship was first used to define and interpret initiatives targeting optimization of antibiotic use. In that time, stewardship gathered momentum to become a movement within healthcare in the fight against AMR. It seems an appropriate moment to revisit what was achieved and what challenges remain in ASP. 

The term stewardship, in the Oxford English Dictionary, is defined as “the job of supervising or taking care of something, such as an organization or a property”. It can also be used in the context of taking care of something that is of value, and it is probably in this context that the term first became associated with antibiotic use: to preserve the valuable efficacy of antibiotic agents. Needless to say, prior to the coinage of this term, initiatives targeting optimization of antibiotic use and the preservation of their efficacy to combat AMR were already in place in healthcare institutions [1,2,8,9]. In the United Kingdom (UK), the British Society for Antimicrobial Chemotherapy (BSAC) had already produced minimum standards for control of antibiotic use in hospitals [9]. The advocacy for optimizing antibiotic use was already on the national and international agenda, with the World Health Organization (WHO) publishing reports on the spread and threat of AMR [10]. In addition, the Copenhagen Declaration [11] had also recommended encouraging good practice in the use of antibiotics through educational initiatives and antibiotic teams. The remit and scope of the Copenhagen Declaration was broad, and, in addition to antibiotic use in community and hospitals, extended to aquaculture, horticulture, and farming. The Copenhagen Declaration remains a landmark report in the fight against AMR, and many subsequent efforts can be traced back to its recommendations. Though it does not refer to “stewardship” as a concept, the Copenhagen Declaration does very clearly define the individual components of what became defined as antibiotic stewardship. Another key attribute of this report was that it included a multidisciplinary approach to its recommendations, through recognizing the role of the infection control specialists, as well as pharmacists, nurses, and the public. The Declaration recommended countries to improve knowledge and attitudes about antimicrobials across all these groups. The Copenhagen Declaration is the only one of the early attempts at an international consensus around rationalizing antibiotic use to put its recommendations in the context of the patient. Specifically, the declaration states that “specialized antimicrobial teams should have the authority to modify antimicrobial prescriptions of individual clinicians, in accordance with locally accepted guidelines, always taking into account of the needs of the patient” [11].

## 2. Tensions between Preserving Antibiotics for the Future and Treating the Present Patient

At its core, the objective of ASP is about preserving the efficacy of antibiotics for use in the future, i.e., the expectation on individual healthcare professionals is to adapt their present behavior and clinical decision-making for a future, intangible reward. This is against the school of teaching in medicine. The modern version of the Hippocratic oath states that doctors should “neither over-treat nor resort to therapeutic nihilism” [12,13]. In the treatment of infections, clinicians often err on the side of over-treatment. The diagnosis and treatment of infections is more complex than many other conditions. The initiation of antibiotics often takes place empirically with minimum information about the pathogen; this can invariably lead to over-treatment. The immediate need of the patient, the central focus of the Hippocratic Oath and referred to in the Copenhagen Declaration, overrides the need to preserve the effectiveness of antibiotics for the future. Therefore, non-compliance with the expectations of an antibiotic stewardship intervention or program is not always tantamount to poor treatment of the patient. It is perhaps these divergent goals of treating the present patient and preserving the efficacy of antibiotics for a future need that make adherence to antibiotic stewardship interventions difficult. Furthermore, the language of antibiotic stewardship and the focal point of all efforts, be it surveillance, antibiotic consumption data, or restrictive formularies, refers to pathogens or antibiotic agents (Table 1). The patient is lost in the midst of all these efforts—this is a key oversight, as due to modern technology and scientific advances, we are increasingly able to treat sicker patients whose therapeutic and clinical needs often drive antibiotic decision-making for individual clinicians. The challenge remains to harmonize the long-term goals of antibiotic stewardship with the short-term goals of clinicians trying to treat infections in sick patients. 

## 3. The International Perspective on Antibiotic Stewardship

Most of the research and evidence for stewardship programs in hospitals continues to be from high-income countries [4,5,16]. The latest Cochrane systematic review of interventions to improve antibiotic prescribing in hospitals included 221 studies from 32 countries [16]. Of the included studies, 183 (83%) were from Europe and North America. The overwhelming majority of published studies are, therefore, from developed healthcare systems and high-income settings, while the overwhelming burden of AMR is in low- and middle-income settings. There is now a growing interest in low- and middle-income countries to implement programs to control and optimize antibiotic use [17,18,19,20]. These efforts are also supported by international reports highlighting the urgent need to tackle AMR on a global scale [21,22]. Research in ASP is necessary from dissimilar cultures, economies, and healthcare organizations. It is only when we know what the challenges are across the spectrum of healthcare globally that we can start finding global solutions. It is often argued that quality in healthcare should be universal, and it should be accountable to the same standards worldwide. Likewise, for ASP, a global standard is necessary. The first step to doing this is to understand what the contextual and cultural determinants are in different countries and how they can be addressed. Many countries are at very different stages of the implementation process for ASP [18,19,23]. To learn to apply the most effective ASP, healthcare organizations need a better understanding of the national and local cultures, and the context in which they work. In a study in Norway, researchers found different factors influencing antibiotic-prescribing behaviors [24]. In Norway, an egalitarian society, hierarchies were less relevant. In the Norwegian context, what took precedent was access to laboratories and workloads, which prevented optimal antibiotic prescribing. Recently, efforts were made to develop core criteria for the implementation of ASP in different healthcare economies, recognizing the role of effective leadership, education, and surveillance as part of sustainable programs [25,26].

## 4. The Role of Culture and Team Dynamics in Antibiotic Stewardship

More recently, a behavioral change framework called the behavior change wheel [27,28] was used in healthcare. This framework is centered on three essential conditions: capability, opportunity, and motivation (COM-B) [29]. This model is based on and adapted from the Theoretical Domains Framework [30] which was developed to map the theoretical constructs for effective implementation research. The Theoretical Domains Framework did include cultural components such as social groups and norms, group conformity, and social support as constructs in its model [30]. Both these models were theoretically developed through a review of literature and expert consensus and included no primary research. The COM-B model was used in research on antibiotic use to understand behaviors and to affect behavior change [31,32]. These published studies, however, did not attempt to provide insight into or address the target population culture and context. Human behavior is a dynamic process of constant interaction between the individual, the physical environment, and the social environment. The behavior change frameworks miss this dynamic by not including culture as a variable in their models. This gap in understanding the culture and context means that such theoretical models may not bring about effective and practical implementation of behavior change in ASP [33].

Today, in hospitals, the patient care pathway includes a multitude of medical and surgical specialties, pharmacists, nurses, and healthcare managers, who are all working to common goals, but, in the process, may prioritize different policies, agendas, and interim targets. Different specialties have their own language, behaviors, social norms, and values [34]. In an ethnographic study seeking to understand the cultural factors that influence antibiotic decision-making across surgical and medical teams, the authors found surgeons to be less willing to tolerate uncertainty, leading to a lower threshold for prescribing antibiotics to patients, particularly in the post-operative period [34]. The cultural differences between specialties and healthcare professionals (1) shape the shared knowledge within and across specialties in the patient pathway, and (2) result in variation in care, thus impacting patient outcomes. The observed variation in care can have serious unintended consequences for patients. By nature, ASPs are driven by multidisciplinary teams [23,35]. From its inception until now, the individuals identified as having a role in ASP are the medical microbiology, infectious disease, specialist pharmacist, and IP&C teams [2,14]. By and large, there is little engagement with other specialties in ASP [36]. Recognizing the importance and contribution of these professions to ASP is critical, and their work contributed to raising the profile of AMR and the need for ASP. However, these individuals are not the target of ASP. Rather, they are the instigators of ASP efforts. The non-infectious disease doctors undertake the bulk of antibiotic prescribing. In hospitals, the diagnostic and treatment pathway of infection in any individual patient is complex and may involve input from multiple teams. Antibiotic prescribing and decision-making is made by multiple individuals for the duration of the infection for any given patient episode. How antibiotic decision-making is made as part of teams outside of the ASP multidisciplinary team is not fully understood or studied. There remains a significant gap in engagement with the greater healthcare workforce whose attitude, knowledge, and behaviors are the target of ASP. As an example, one study investigating cross-specialty clinical engagement on ASP found it to be extremely poor with very little coverage of AMR and ASP across intensive care, surgical, and general medical conferences [37]. 

Most of the published ASP studies report an intervention initiated by members of the traditional ASP teams [16]. Research is needed to investigate how antibiotic decision-making takes place in the absence of ASP specialists to investigate what determines decisions around antibiotic use in daily routine care of acute medical and surgical wards. This is critical to developing effective ASP programs. To change existing behaviors, one must first understand the social norms that drive and determine antibiotic decision-making. To help understand this, a social science perspective is required. We are social animals who interact and moderate our behaviors as result of the response we get from our peers and social counterparts. In healthcare, there are many different affiliations and groups to which an individual can belong. There are inter-professional groups, e.g., doctors, pharmacist, and nurses, and intra-professional groups, e.g., surgery, medicine, hematology, respiratory medicine, etc. In short, we work or are expected to work as part of teams.

The term culture is defined and interpreted in many ways [38,39,40]. Spradley defined culture to mean “the acquired knowledge people use to interpret, experience, and generate behavior [41]. “In this context culture refers to how people acquire knowledge and moderate their behaviors as members of a group [41]. Cultural knowledge does not and should not underpin every action that individuals take; rather, it acts as an internal barometer, which is used to interpret and evaluate situations. Culture is a “cognitive map to which individuals refer to when making decisions” [41,42]. In healthcare, there are many different meso and macro cultures, at the organizational, specialty, and professional level [43,44]. Indeed, terms such as “safety culture” [44,45] and “no-blame culture” [46,47] are used extensively to try and promote and reinforce desired behaviors amongst healthcare professionals. Interventions are designed, implemented, and expected to be adopted with little thought given to understanding the culture and context in which they are to function sustainably. In research published in 2004, using ethnographic methods, Gabbay and colleagues investigated general practitioner use of evidence-based guidelines. The study found that, rather than referring to evidence-based guidelines, clinicians use tacit sources of knowledge which the authors called “mindlines” [48]. These mindlines consisted chiefly of interactions with opinion leaders, patients, and their own and their colleagues’ experiences [48], i.e., culture. This study, though conducted across primary care, represents an early example of conducting qualitative research in order to understand the influence of context on clinical decision-making. In secondary care, culture is also a key determinant of behaviors. In a systematic review published in 2016, Braithwaite and colleagues identified a positive organizational culture (defined as values, norms, and beliefs) with positive patient outcomes [49]. The decline effect [50], where promising effects of an intervention are difficult to replicate in different settings, is a prime example of the influence of culture and context on behavioral outcomes. This was demonstrated in an ethnographic study conducted across 200 intensive care units in England that had implemented a program to reduce central line infections [51]. Successful units were identified as those where efforts had been made to understand the context in which the interventions were to be implemented. This included flattening hierarchies where they existed, championing local leadership, and involving healthcare professionals in the decision-making process. In unsuccessful units, efforts had not been made to engage with and persuade senior consultants to adopt the program [51]. This study highlights the necessity to understand and address local cultures and context when developing interventions aiming to change healthcare professional behaviors. Whether tacit or explicit, culture matters in healthcare. It has the power to moderate behaviors and shape intervention outcomes. We need a better understanding of culture if we want to develop interventions which are successful and sustainable. This is particularly true in ASP, where multiple teams share responsibility for the treatment of infections in individual patients. Previous research highlighted the existence of professional hierarchies and etiquette as key determinants of antibiotic decision-making in hospitals [48,52,53]. In a study mapping antibiotic decision-making of medical specialists, it was reported that deciding the pathway for infection management in medicine is systematic; however, the junior doctors felt left out of the decision-making [54]. In surgery, the culture of individualism and disjointed communication means that antibiotic decision-making can often remain unowned [34,55]. The surgical teams operate to different priorities, and there is a willingness to delegate antibiotic decision-making to other colleagues, for example, from the medical specialties. Understanding the contextual drivers of antibiotic decision-making will provide a valuable opportunity for shared learning for design and implementation of bespoke ASP, which attends to the specialty level variations in practice [56]. Much of the evidence for ASP in the surgical pathway focuses on surgical prophylaxis and surgical site infection prevention. There are opportunities to develop bespoke ASP that targets antibiotic decision-making at each step of the surgical pathway [57].

## 5. Conclusions

In the last 20 years, antibiotic stewardship evolved continuously, and is now a global drive, with organizations aiming to implement interventions to rationalize antibiotic use in secondary care. To influence behaviors of prescribers, frameworks from social science research are being used. However, these frameworks often do not include a study of context and culture. To bring about sustainable change in prescribing behaviors, and to optimize antibiotic use, it is first necessary to study how and why healthcare professionals behave the way they do. To do this, it is necessary to undertake research into the cultural determinates of antibiotic use in secondary care.

## Figures and Tables

**Table 1 antibiotics-08-00007-t001:** The evolving definition of antibiotic stewardship.

1997 Society for Healthcare Epidemiology of America and Infectious Diseases Society of America [2]	2014 Center for Disease Control [14,15]
**Appropriate antimicrobial stewardship includes optimal selection, dose, and duration of treatment, as well as control of antibiotic use.** **Statement on antibiotic stewardship programs (ASPs)** **The ideal is to have all patients treated with the most effective, least toxic, and least costly antibiotic for the precise duration of time to cure or prevent an infection.** **The key components** Precise definitions of antimicrobial resistance for antimicrobials and organisms: A system for monitoring the frequency of resistance (clinical and environmental); A determination of which antimicrobial(s) to control; A method to achieve usage control; A determination of who will be responsible for maintaining control; A method to educate and enroll prescribers in the control process; A stable system of hospital infection control; A system to measure use of controlled and uncontrolled antimicrobials; A method to determine antimicrobial use per geographic area per unit time; Ability to distinguish community from nosocomial isolates; Ability to identify isolates by body site and hospital location; A method to assure that clinical care will not be harmed by control measures; Ability to identify known mechanisms of antimicrobial resistance.	**Antimicrobial stewardship programs can increase the frequency of appropriate prescribing, optimize the treatment of infections, and minimize adverse events associated with antibiotic use, including Clostridium difficile infections (CDIs).** **Statement on antibiotic stewardship programs (ASP)** **Strategies for improving antibiotic use and evidence for best practices in antibiotic stewardship are evolving.** **The key components** Leadership commitment: Dedicating necessary human, financial, and information technology resources. Accountability: Appointing a single leader responsible for program outcomes. Experience with successful programs shows that a physician leader is effective. Drug expertise: Appointing a single pharmacist leader responsible for working to improve antibiotic use. Action: Implementing at least one recommended action, such as systemic evaluation of ongoing treatment need after a set period of initial treatment (i.e., “antibiotic time-out” after 48 h). Tracking: Monitoring antibiotic prescribing and resistance patterns. Reporting: Regular reporting information on antibiotic use and resistance to doctors, nurses, and relevant staff. Education: Educating clinicians about resistance and optimal prescribing.

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
