# Peer review of "Antibiotic Stewardship—Twenty Years in the Making"

_antibiotics, 2019, doi:10.3390/antibiotics8010007_

Reviewer 1 Report

Dear authors

This is a very important and relevant article summarising in a succinct way achievements and challenges in the AMS. It also reads very well.

I have a few minor comments.

The Introduction makes it very clear that the main focus of the paper is secondary care; this also comes across in the section “Role of Culture”. This focus is somehow not as clear in the section “The international perspective on antibiotic stewardship”. Perhaps it could be made clearer what settings you are referring to here.

In the section: “ the role of culture”, you mention one study on primary care; this was a bit unexpected as the whole section is mainly about challenges in secondary care?

Author Response

Reviewer:

This is a very important and relevant article summarising in a succinct way achievements and challenges in the AMS. It also reads

Response:

We thank the reviewer for acknowledging the importance of this article.

Reviewer:

I have a few minor comments.

The Introduction makes it very clear that the main focus of the paper is secondary care; this also comes across in the section “Role of Culture”. This focus is somehow not as clear in the section “The international perspective on antibiotic stewardship”. Perhaps it could be made clearer what settings you are referring to here.

Response:

We have amended the first sentences in this section to: ‘Most of the research and evidence for stewardship programmes in hospitals continues to be from high income countries(4,5,16). The latest Cochrane systematic review of interventions to improve antibiotic prescribing in hospitals, included  221 studies from 32 countries(16).’

Reviewer:

In the section: “ the role of culture”, you mention one study on primary care; this was a bit unexpected as the whole section is mainly about challenges in secondary care?

Response:

 We have added the following sentence to clarify why we included this study in this review: ‘This study, though conducted across primary care, represents an early example of conducting qualitative research in order to understand the influence of context on clinical decision-making.’

Reviewer 2 Report

-pg 4, first 5 lines of paragraph.  Authors state that there are a number of studies on stewardship programs conducted in Europe and North America.  Although referenced, it would help to clarify this by providing an example of such a study and what the results were.  In other words, what characteristics distinguishes studies conducted in developed countries?

-pg 4, last paragraph discussed that there are a variety of healthcare specialists involved in patient care and that each of these specialists have different priorities based on their experiences which could affect the use of antibiotics.  It would be helpful to provide at least one example of how this can happen and the results of antibiotic use on patient care.

-pg6, top paragraph.  Other groups of professionals based on hierarchy (seniority) can affect antibiotic decision-making processes.  The conclusion arrived at implied that junior level physicians are left out.  Is there any specific data from a study or studies that support this?

Author Response

Reviewer:

pg 4, first 5 lines of paragraph.  Authors state that there are a number of studies on stewardship programs conducted in Europe and North America.  Although referenced, it would help to clarify this by providing an example of such a study and what the results were.  In other words, what characteristics distinguishes studies conducted in developed countries?

 Response:

 The point we are trying to make here is that the majority of burden of infectious diseases and AMR remain in low and middle income countries, whilst the majority of the research in stewardship is from developed settings. We have edited the sentence to reflect this:

‘The overwhelming majority of published studies are therefore from developed healthcare systems and high income settings, whilst the overwhelming burden of AMR is in low and middle income settings.’

Reviewer:

-pg 4, last paragraph discussed that there are a variety of healthcare specialists involved in patient care and that each of these specialists have different priorities based on their experiences which could affect the use of antibiotics.  It would be helpful to provide at least one example of how this can happen and the results of antibiotic use on patient care.

Response:

We have added the following:

Different specialties have their own language, behaviours, social norms, and values(34). In an ethnographic study seeking to understand the cultural factors that influence antibiotic-decision-making across surgical and medical teams, the authors found surgeons to be less willing to tolerate uncertainty leading to a lower threshold for prescribing antibiotics to patients, particularly in the post-operative period(34).

Reviewer:

-pg6, top paragraph.  Other groups of professionals based on hierarchy (seniority) can affect antibiotic decision-making processes.  The conclusion arrived at implied that junior level physicians are left out.  Is there any specific data from a study or studies that support this?

Response:

This sentence based on findings from the study in reference 54, this sentence already included in the manuscript refers to the study:

In a study mapping antibiotic decision making of medical specialists it was reported that decision the pathway for infection management in medicine is systematic, however the junior doctors felt left out of the decision making(54).’
